# Formation of Hydrophilic Nanofibers from Nanostructural Design in the Co-Encapsulation of Celecoxib through Electrospinning

**DOI:** 10.3390/pharmaceutics15030730

**Published:** 2023-02-22

**Authors:** Kedi Chu, Yi Zhu, Geng Lu, Sa Huang, Chuangzan Yang, Juying Zheng, Junming Chen, Junfeng Ban, Huanhuan Jia, Zhufen Lu

**Affiliations:** 1Center for New Drug Research and Development, Guangdong Pharmaceutical University, Guangzhou 510006, China; 2The Innovation Team for Integrating Pharmacy with Entrepreneurship, Guangdong Pharmaceutical University, Guangzhou 510006, China; 3Guangdong Laboratory Animals Monitoring Institute, Guangdong Provincial Key Laboratory of Laboratory Animals, Guangzhou 510663, China; 4Guangdong Provincial Key Laboratory of Advanced Drug Delivery Systems, Guangdong Pharmaceutical University, Guangzhou 510006, China; 5Guangdong Provincial Engineering Center of Topical Precision Drug Delivery System, Center for Drug Research and Development, Guangdong Pharmaceutical University, Guangzhou 510006, China

**Keywords:** nanoparticle, nanofiber, drug delivery, osteoarthritis

## Abstract

This study presents a method for a one-step co-encapsulation of PLGA nanoparticles in hydrophilic nanofibers. The aim is to effectively deliver the drug to the lesion site and achieve a longer release time. The celecoxib nanofiber membrane (Cel-NPs-NFs) was prepared by emulsion solvent evaporation and electrospinning with celecoxib as a model drug. By this method, nanodroplets of celecoxib PLGA are entrapped within polymer nanofibers during an electrospinning process. Moreover, Cel-NPs-NFs exhibited good mechanical strength and hydrophilicity, with a cumulative release of 67.74% for seven days, and the cell uptake at 0.5 h was 2.7 times higher than that of pure nanoparticles. Furthermore, pathological sections of the joint exhibited an apparent therapeutic effect on rat OA, and the drug was delivered effectively. According to the results, this solid matrix containing nanodroplets or nanoparticles could use hydrophilic materials as carriers to prolong drug release time.

## 1. Introduction

With the aging of the population and the increasing obesity rate, the prevalence of osteoarthritis (OA) is increasing. The course of OA is long and the disability rate is high. Joint replacement is often needed in the end stage, which causes pain and reduces patients’ quality of life [1,2]. The pathological changes caused by OA involve the whole joint, and local intra-articular administration can distribute a drug through the whole joint capsule, which is an ideal way of drug administration [3,4]. Drugs are removed mainly through capillaries or lymphatic pathways in joints. Small molecules with a relative molecular weight of <10 × 10^3^ can generally be absorbed into the circulatory system through capillaries, while macromolecules are cleared through lymphatic pathways [5]. In addition, this phenomenon will change with the development of OA [6]. Prolonging the residence time of the drug in the joint with a low dose and low-frequency of injection is very important for the curative effect of OA [7]. The difficulty of intra-articular drug delivery can be overcome by using nanomaterials to deliver drugs.

On the one hand, nanomaterials increase the relative molecular weight of free drugs, prolonging the residence time of drugs in joints. Nanoscale carriers can penetrate the extracellular matrix (ECM) and cell membrane, and release drugs directly into the ECM or into cells [8]. Non-steroidal anti-inflammatory drugs (NSAIDs) are quickly eliminated from the articular cavity [9], with the half-life in the articular cavity being only a quarter of an hour [6]. By targeting the joint tissue, we can effectively deliver the drug to the expected site, reduce the dosage and side effects of the drug, increase the residence time of the drug in the target tissue and the whole joint, and enhance the curative effect. 

In this study, a drug delivery system using polylactic acid-hydroxyacetic acid copolymer (PLGA) nanoparticles and polyvinyl alcohol (PVA) was developed for intra-articular implantation of celecoxib nanoparticle fibrous membranes (Cel-NPs-NFs) by electrostatic spinning. The system was first prepared using PLGA and 2-HP-βCD as carriers by emulsifying solvent volatilization to produce drug-loaded nanoparticles (Cel-NPs). Next, the nanoparticles were mixed with a polymer solution and then encapsulated in nanofibers using electrostatic spinning technology. During this process the polymer solution mixed with nanoparticles is injected through a needle to produce a highly expanded and curved fluid jet and the dispersed phase tends to collect in the center of the fluid. The elongation effect of the fluid in the air helps the nanoparticles to settle inside the fibers rather than on the surface [10,11]. This results in a nanofiber film with most of the drug-laden nanoparticles uniformly dispersed inside the fiber, as shown in Figure 1. This paper focuses on verifying the feasibility of this drug delivery system, characterizing its physicochemical properties, assessing its cell uptake in the RAW264.7 cell model in vitro, and verifying the function of the drug delivery system in the SD rat knee joint OA model in vivo.

## 2. Materials and Methods

### 2.1. Materials

Polyvinyl alcohol (PVA-205, PVA-1788) was purchased from Corey International Trading Shanghai Co., Ltd. (Shanghai, China); 2-hydroxypropyl-β-cyclodextrin (2-HP-β-CD) was obtained from Zibo Qianhui Biotechnology Co., Ltd. (Shandong, China); Celecoxib was supplied by Tixiai Shanghai Huacheng Industrial Development Co., Ltd. (Shanghai, China); Poly (lactic-co-glycolic acid) (PLGA) supplied by Jinan Jufukai Biotechnology Co., Ltd. (Shandong, China); DMEM, FBS, and PBS (pH = 7.4) buffer for cells was purchased from Gibco (California, CA, USA); CCK8 enhanced solution supplied by Meilun Biotechnology Co., Ltd. (Liaoning, China); Coumarin 6 was from Aladdin Reagent Shanghai Co., Ltd. (Shanghai, China); DAPI Solution (1 mg/mL), dimethyl sulfoxide (cell culture grade) and trypsin-EDTA digest solution were obtained from Beijing Solarbio Science & Technology Co., Ltd. (Beijing, China).

### 2.2. Preparation of Celecoxib Hydrophilic Nanofibers

#### 2.2.1. Nanostructural Design in Co-Encapsulation of Celecoxib

Celecoxib nanoparticles were prepared and optimized based on the emulsified solvent evaporation method [12]. Briefly, 10 mg of celecoxib was mixed with 50 mg of PLGA (*w*:*w*, 1:5) and dissolved in 5 mL of acetone and dichloromethane mixed solution (*v*:*v*, 3:2) to form phase I. Then 20 mL of 1.5% HP-β-CD and 2% PVA aqueous solution formed phase II. Then phase I was injected slowly into phase II at 90 W ultrasonic power, 20kHz frequency, and phacoemulsification interval of 3 s for 30 min (JY88IIN, Ningbo, China), followed by stirring at a low speed of 200 rpm (IKA RCT, Staufen, Germany) until the phase I organic solvent evaporated completely. Finally, distilled water was added until a volume of 20 mL was reached to obtain a 0.05% celecoxib nanoparticle solution (Cel-NPs).

#### 2.2.2. Electrospinning

Using PVA as a membrane material, a celecoxib nanoparticle fiber membrane was prepared by electrospinning. First, 9% (*w*/*v*) PVA film-forming material (hydrolysis degree: 88%, polymerization degree: 1700:500; 3:1, *w*/*w*) was added to the drug-containing nanoparticle solution until it was fully swollen and dissolved. Then the spinning solution was obtained. Next, with a voltage of 20 kV, injection rate of 0.7 mL/h, and a receiving distance of 140 mm, the uniaxial method (27G needle, outer diameter: 0.41 mm, inner diameter: 0.20 mm) was adopted to spin the spinning solution with a high-voltage electrostatic spinning machine (NAN0N-01A, MECC, Tokyo, Japan) and finally, a 3 × 5 mm film (Cel-NPs-NFs) was cut from the obtained drug-containing membrane.

### 2.3. Nanocarrier Size, Polydispersity, and Particle Charge

The particle size distribution, PDI, and Zeta potential characteristics of the Cel-NPs and Cel-NPs in Cel-NPs-NFs were determined by Delsa Nano C /Zeta potential analyzer (Beckman, California, CA, USA). Cel-NPs were directly diluted 10 times with water, and the emission scattering intensity was adjusted to (10,500 ± 1500) cps. Cel-NPs in Cel-NPs-NFs were measured by taking 0.1 g of Cel-NPs-NFs and dissolving it in 2 mL of water to prepare the sample solution; then, the emission scattering intensity was adjusted to (10,500 ± 1500) cps for measurement.

### 2.4. Shape and Morphology

The microscopic characterization of Cel-NPs and Cel-NPs-NFs was performed by transmission electron microscopy (Talos L120C, Thermo Fisher, New York, NY, USA). Cel-NPs and Cel-NPs in Cel-NPs-NFs underwent a double dyeing method [13]. Cel-NPs solution was diluted ten times, and 0.1 g of Cel-NPs-NFs was dissolved in water and diluted to an appropriate multiple. A sample solution was transferred onto the copper net covered with carbon film and stained with a 2% phosphotungstic acid solution. After the copper net dried, the surface morphology of Cel-NPs was observed under an electron microscope. The spinning solution was directly spun on the copper net, and the surface morphology was observed after the copper net was dried. The prepared sample was cut to an appropriate size, attached to the copper strip, and sprayed with gold at 20 mV for 20 s. The internal structure of Cel-NPs-NFs was observed under a scanning electron microscope (SEM, FEI Nova NaoSEM 450, New York, NY, USA). Image J (1.52P, Wayne Rasband National Institutes of Health, New York, NY, USA) software was used to analyze the distribution of fiber diameters in Cel-NPs-NFs.

### 2.5. Encapsulation Efficiency

The encapsulation rate of celecoxib in electrospun co-encapsulation of celecoxib NPs was determined by a high performance liquid chromatograph (HPLC, e2695, Waters, Milford, CT, USA) ultrafiltration-assisted low-speed centrifugal method [14,15]. The operation process was as follows: Firstly, the Cel-NPs solution (C_3_) underwent low-speed centrifugation at 1000 rpm for 10 min, and Cel-NPs and a small amount of free drug was separated. Then, the free drug was deposited at the bottom; the upper liquid was a mixture of Cel-NPs and drug solution (C_1_). Secondly, the upper liquid was transferred into an ultrafiltration tube (molecular weight cutoff: 3k) and centrifuged at 4 °C and 13,000 rpm (TDL-60B, Shanghai, China) for 30 min. Then, a small amount of free drug (C_2_) and Cel-NPs were separated. Finally, the encapsulation rate (EE%) of celecoxib in electrospun co-encapsulation of celecoxib NPs was determined according to Equation (1). 

The content of celecoxib in Cel-NPs solution was determined by HPLC (e2695 Waters HPLC with a Kromasil 100-5-C18 column (4.6 × 250 mm)), with mobile phase: the methanol-to-water ratio was 85:15, the injection volume was 20 μL, the flow rate was 1 mL/min, the column temperature was kept at 25 °C, and the detection wavelength was 254 nm). The Cel-NPs solution of 1 mL was transferred to a 10 mL volumetric flask, and an appropriate amount of methanol was added. After ultrasonic de-emulsification at 53 KHz for 30 min, the solution was filtered through a 0.22 µm microporous filter membrane, and the drug content was determined by HPLC.
(1)$$\mathrm{EE}\%=\frac{C_{1}{-C}_{2}}{C_{3}}\times 100\%$$

### 2.6. Determination of Drug Encapsulation Form in Nanofiber Membrane

#### 2.6.1. Fourier Transform Infrared Spectroscopy (FTIR)

IR (Nicolet iS5, Thermo Fisher, New York, NY, USA) was used to distinguish the mode of action between celecoxib and carrier by taking appropriate amounts of celecoxib, Cel-NPs, and Cel-NPs-NFs. In the IR test, the dried sample and potassium bromide were uniformly ground evenly according to mass ratio of 1:100. Then, the tablets were scanned and analyzed in the wavelength range of 400–4000 cm^−1^ at room temperature.

#### 2.6.2. Differential Scanning Calorimetry (DSC) 

DSC (DSC4000, PerkinElmer, Milford, CT, USA) was used to analyze the presence of the drug in the formulation. Then, 5 mg of celecoxib, Cel-NPs and Cel-NPs-NFs samples were weighed. The detection conditions of DSC were as follows: the heating rate was 10 °C /min, and the temperature range of the instrument was 30–300 °C.

#### 2.6.3. X-ray Diffraction (XRD) 

XRD (Rigaku-SmartLab SE, Tokyo, Japan) was used to further analyze the mode of action of celecoxib with the carrier. Celecoxib and Cel-NPs test state was powder, and Cel-NPs-NFs test state was a thin film. The parameters of the XRD measurement were as follows: the diffraction angle was 460°, the scanning step was 0.02°, the scanning mode was 2θ, the voltage of the Cu tube was 40 kV, and the current was 40 mA.

### 2.7. Performance Test of Nanofiber Membrane

Electrospun nanofibers have the advantages of a small diameter and a large specific surface area, and different membrane materials and drug loading methods affect the drug release rate at the drug delivery site. In this experiment, the membrane’s contact angle and mechanical properties were established to evaluate the process that could prepare the membrane. The contact angle can reflect changes in the wettability of the film surface: the smaller the contact angle, the better the wettability [16]. The surface wettability of Cel-NPs-NFs was analyzed by a contact angle tester (SDC-100, DingSheng, Dongguan, China). The sample was cut to a suitable size and placed on the table of the contact angle tester. Deionized water droplets were placed on the sample surface using a micro-syringe with a flat needle. The droplet shape image was collected when the droplet touched the surface, and the instantaneous contact angle was measured by Image J (1.52 P Wayne Rasband National Institutes of Health, New York, NY, USA) software. The tensile test method of the film texture tester (TA-XTplus, StableMicroSystems, London, UK) was established [17]. Using an A/TG probe and a 500 N sensor, the sample was cut to 50 × 10 mm, the distance between clamping points was 30 mm, and a length of 10 mm was clamped up and down. The stress–strain curve was measured at the tensile speed of 200 mm/min. Then, Young’s modulus, fracture strength, and elongation were obtained. The deformation process, deformation capacity, and deformation degree of Cel-NPs-NFs were evaluated.

### 2.8. In Vitro Celecoxib Release Study

The in vitro release behavior of Cel-NPs and Cel-NPs-NFs were evaluated by the dialysis bag method [18,19,20], simulating the synovial fluid microenvironment in vivo. Firstly, different preparations (celecoxib content: 80 µg) and an appropriate amount of medium were added to the dialysis bag (molecular weight cutoff of 8–14 kDa), fastened at both ends, and placed in a 50-mL centrifuge tube containing 25 mL of 0.5% SDS phosphate-buffered release medium. Secondly, they were placed in a constant-temperature shaker (IS-RSV3, JieMei, SuZhou, China) at 37 °C and 100 rpm, respectively, for 1, 3, 6, 12, 24, 72, 96, and 120 h. At 168 h, the 2-mL release medium was removed, and the same volume of release medium was added. Finally, the drug content was determined by HPLC. The column temperature was kept at 25 °C, and the detection wavelength was 254 nm after the release solution was filtered. Then, the in vitro release curve was calculated and drawn according to Equation (2).
(2)$$X=\frac{{(C}_{t}\;\times \;25+V\sum_{i=1}^{t-1}C_{i})}{m_{0}}\;\times \;100\%$$X is the cumulative drug release at t time point, m_0_ is the total amount of drug injected and released, C_t_ is the drug concentration at time point t (ug/mL), C_i_ is the drug concentration measured at a time point before the t time point (ug/mL), V is the sampling volume (2 mL).

### 2.9. Evaluation of Cellular Uptake Efficiency

Coumarin is a substance with similar solubility to celecoxib and can be used as a probe for fluorescence labeling. The nanoparticle fiber membrane (C6-NPs-NFs) co-encapsulating coumarin-6 (C6-NPs) was prepared by the same method, and its particle size potential was determined by nanocarrier size, polydispersity, and particle charge methods to study the uptake of the preparation in mouse macrophage RAW 264.7 (the cell line used was derived from Abelson murine virus-induced leukemia and was purchased from Suzhou Meilun Biotechnology Co., Ltd. (Suzhou, China) Cell item number was PWE-MU004-2). The CCK8 method was used to determine the cytotoxicity of blank nano-carrier solution and nano-fiber spinning solution when the concentration of celecoxib was a gradient of 125–15,625 µg/mL. RAW264.7 was cultured in a 12-well plate (1 × 10^5^ cells per well) for 24 h while the cells grew to 80%. C6-NPs and C6-Nps-NFs containing 0.18 µg coumarins were cultured in serum-free medium and incubated for 0.5 h and 4 h, respectively, without light at 37 °C. The supernatant was discarded, and the cells were fixed in 4% paraformaldehyde solution at room temperature for 15 min, washed with cold PBS three times, and then observed under a fluorescence confocal microscope (STELLARIS, Leica, Wetzlar, Germany). The density of 2 × 10^5^ cells per well was quantitatively analyzed. After the above steps, until the end of uptake and trypsin digestion, the cells were collected by 3 × 10^3^ r/min, centrifuged for 5 min, and resuscitated with 0.5 mL PBS. The average fluorescence intensity (MFI) of 10,000 cells was detected by flow cytometry (BD/FACSCanto2, BD, Illinois, IL, USA).

### 2.10. In Vivo Safety and Effect Study

The animal model of osteoarthritis (OA) established by injecting sodium iodoacetate into the knee joint is a classical and reliable model to study osteoarthritis [21,22]. The pathological manifestations, such as joint pain, intermittent inflammation, and joint nerve injury, are similar to the clinical symptoms of arthritis patients. Considering the apparent inhibitory effect of celecoxib (20 µg) on cartilage injury caused by inflammation [23], the effect of electrospun fiber membrane on arthritis was quantitatively evaluated in rat knees to explore the pathological effect of the preparation on synovium and cartilage. The Institutional Animal Care and Use Committee of Guangdong Pharmaceutical University approved the study, ensuring that the care and use of animals conformed to the National Institutes of Health’s guide for the care and use of laboratory animals. The procedure was as follows: 2% pentobarbital sodium (40 mg/kg) intraperitoneal injection anesthesia, knee joint bend of 45°, joint cavity puncture from the outside of the inferior patellar ligament, one-time injection of 50 µL of 4% sodium iodoacetate solution (0.9% normal saline preparation, the concentration was 2 µg/50 µL). The rat knee was stretched and bent for 30 s so that the sodium iodoacetate solution was dispersed through the entire knee joint. All the rats were fed similarly, and their behaviors, such as hind limb tension, were scored by Lequesne MG [24]. They were randomly divided into the model group, sham operation group, positive control group, and Cel-NPs-NFs treatment group (*n* = 8). The positive control group was given 10.5 mg/kg celecoxib capsules by intragastric administration once a day. In contrast, in the Cel-NPs-NFs treatment group, the nanoparticles were implanted in the articular cavity (drug content: 20 µg). Seven days later, the rats were intraperitoneally injected with 2% pentobarbital sodium and sacrificed. The skin and subcutaneous tissue were separated from the knee joint, and the surrounding muscle tissue was removed. The whole knee joint of rats was amputated, preserving a quarter of the femur and tibia. The injury to cartilage was observed by photography, and the gross cartilage was scored by the Pelletier scoring standard [25]. The knee joint was preserved in 4% paraformaldehyde, then decalcified with 5% EDTA, embedded in paraffin, sliced, and stained with hematoxylin-eosin and saffron green. The pathological changes of the cartilage were observed under a microscope and evaluated according to the Mankin score standard [26].

### 2.11. Statistical Analysis

The data were collected and analyzed. GraphPad prism 8.0.2 was used for statistical analysis of the experimental data and expressed by X ± SD. Pairwise comparisons were made by the Welch-t test. Dunnett’s multiple comparison tests were used for comparison between multiple groups. The tests revealed no significant differences when *p* > 0.05.

## 3. Results and Discussion

### 3.1. Preparation and Characterization of Cel-NPs

Nanoparticles can increase the amount of drugs delivered into inflammatory synovial tissues and are more advantageous than microspheres and other particle drug delivery systems [27]. The copolymerization ratio and molecular weight properties of PLGA are the key factors affecting the formation of nanoparticles. In the experiment, the molecular weight of PLGA was 1 w, and the differences of D90 under different copolymerization ratios (50:50, 75:25, and 90:10) were compared. As shown in Figure 2a, D90 was the smallest when the copolymerization ratio of PLGA was 90:10; therefore, the copolymerization ratio of PLGA was 90:10 to achieve a smaller particle size. When the copolymerization ratio was 90:10, the average particle size was used as the evaluation index to investigate the effect of different molecular weights (10, 23, and 50 kDa) on the system. As shown in Figure 2b, the larger the molecular weight, the larger the average particle size. The NPs prepared on this basis were blue, clear, transparent, and stable (Figure 2c). The average particle size distribution was 74.23 ± 8.22 nm (Figure 2e), the polydispersion index was 0.175 ± 0.058, and the potential was −38.58 ± 5.64 mV (Figure 2f). The nanoparticles exhibited a spherical structure (Figure 2d). The nanoparticles were uniformly distributed, the particle size distribution was narrow, and the system was stable without precipitation and condensation.

### 3.2. Characteristics of Co-Encapsulation Obtained by Nanoemulsion Structural Design

In this study, celecoxib was first prepared as Cel-NPs. Then Cel-NPs-NFs were prepared, combining the advantages of NFs and NPs, to provide a reference for implanted drug administration of arthritis. There was no apparent change in the morphology of Cel-NPs before and after electrospinning. Cel-NPs before and after electrospinning exhibited a spherical shape (Figure 3a). The average particle size was 72.53 ± 2.27 nm (Figure 3b), the potential was −58.74 ± 16.58 mV (Figure 3c), and the polydispersion index was 0.219 ± 0.019. The prepared Cel-NPs-NFs were dense in appearance and had a certain supporting effect (Figure 3d). The surface of the fiber was smooth and uniform under the transmission electron microscope (Figure 3e) because the hydrophilic PVA polymer inhibited the aggregation of Cel-NPs on the nanofibers. Therefore, Cel-NPs-NFs did not stack on the fibers but were evenly distributed inside the fiber [17]. Under the scanning electron microscope, the diameter distribution was measured by collecting Cel-NPs-NFs 100 roots as follows (Figure 3f), and the average diameter was 245.22 ± 27.58 nm.

Through the tensile test, we can judge the change in the stress–strain curve of the film in the elastic stage, yield stage, strengthening stage, and local deformation stage (Figure 3g). The Young’s modulus of Cel-NPs-NFs was 188.90 ± 52.49 MPa, the breaking force was 11.42 ± 1.18 MPa, and the elongation at break was 17.27 ± 3.51%. There was no yield point in the stress–strain curve at different stages. Fracture after no yield phenomenon directly reaches the fracture force. The contact angle is the key factor affecting the contact between the film and the administration site. The contact angle of Cel-NPs-NFs was measured. The results showed that the instantaneous contact angle of the film was <90° (70.87 ± 1.43)° and had hydrophilicity.

### 3.3. Analysis of Drug Distribution

As shown in Table 1, the entrapment efficiency of celecoxib in electrospun co-encapsulation of celecoxib NPs was 88.23 ± 4.61% by low-speed centrifugation and ultrafiltration. The drug content was 1.11% in Cel-NPs and 0.38% in Cel-NPs-NFs. 

The stretching vibration characteristic double peak of primary amine N-H at 3237 cm^−1^ and 3342 cm^−1^ is the characteristic peak of celecoxib. The characteristic peak of S=O_2_ could be seen at 1402 cm^−1^ and 1146 cm^−1^ (Figure 4a) [28]. These peaks disappeared or decreased in the molding preparations of Cel-NPs and Cel-NPs-NFs (Figure 4a) due to the encapsulation effect. 3400 cm^−1^ is the -OH vibration absorption peak of PVA, 2916 cm^−1^ is the asymmetric absorption vibration peak of CH_2_, 1642 cm^−1^ is the vibration absorption peak of C=O, and 1089 cm^−1^ is the stretching vibration peak of C-O [17,29]. PVA is composed of many vinyl alcohol segments, with many hydroxyl groups, intermolecular hydrogen bonds, and terminal unsaturated olefins in the molecular structure. There is a bending vibration absorption peak of C-C at 858 cm^−1^; however, the peak area is small. Therefore, it is not easy to produce multi-point infrared absorption peaks, so PVA has little effect on determining infrared transmittance [30]. The disappearance of the characteristic peak showed that celecoxib was well encapsulated in NPs and Cel-NPs existed in NFs in complete form. Celecoxib produced a phase transition endothermic peak at 166.52 (Figure 4b) [28]. The broad peak of Cel-NPs at 30–100 was formed by water evaporation, and PVA produced two endothermic decomposition peaks at 55.69 and 197.83 [29,31]. The endothermic decomposition peak of PVA was produced at the same position in Cel-NPs-NFs, but the phase transition endothermic peak of celecoxib was not observed in Cel-NPs-NFs and Cel-NPs (Figure 4b), indicating that celecoxib in NPs-NFs was in an amorphous state. The XRD curve can also be proved. The main diffraction peaks of celecoxib are 5.4°, 10.8°, 13.1°, 14.9°, 16.1°, 17.9°, 19.7°, 21.5°, 22.4°, 23.6°, 25.4°, 29.5° and so on (Figure 4c), indicating that celecoxib is in a crystalline state. In the physical mixture of Cel-NPs-NFs, due to the existence of high molecular weight PVA with high crystallinity, there is a strong diffraction peak at 19.8°and a weak diffraction peak at 40.6° (Figure 4c), while the peak area decreases and the peak at 40.6° disappears and the characteristic peak of celecoxib disappears in the Cel-NPs-NFs molding preparation. This phenomenon shows that the overall crystallinity of the preparation decreases, and the drug exists in the carrier in an amorphous state which interacts with the carrier in the process of preparation, and the crystallization state changes.

### 3.4. Analysis of In Vitro Drug Release by Co-Encapsulation Cel-NPs-NFs

As shown in Figure 5, the cumulative release of Cel-NPs was 25.78% and 43.56% in the first 1 h and 3 h, respectively. The sudden release of celecoxib in the first 3 h may be attributed to the adsorption of some celecoxib on the Cel-NPs bound to the carrier due to solvent volatilization, resulting in a higher drug concentration on the surface of NPs. After contacting the release medium, the rapid distribution of the easy-binding drug on the surface to the release medium is related to structural relaxation [32]. Subsequently, due to the swelling and degradation of PLGA, the drug was released slowly up to 72 h, and the cumulative release reached 82.25%. The uncontrollable release behavior caused by this sudden release behavior of NPs will significantly impact its application, while in Cel-NPs-NFs, due to the role of PVA, the binding of NPs is closer, its integrity is maintained better, and its structural relaxation is restrained. The release curve also showed that Cel-NPs-NFs increased the drug diffusion resistance and prevented the sudden release behavior of NPs. The cumulative release of Cel-NPs-NFs at 168 h was 67.74%, indicating a long-term release effect. The kinetic models of the two nanoparticle types were fitted, respectively. The fitting results (Table 2) show that Cel-NPs and Cel-NPs-NFs are close to the Ritger-Peppas release model, except that the n value of Cel-NPs is 0.1841, which is <0.5, and the release process is mainly controlled by Fick diffusion, and the n value of Cel-NPs-NFs is 0.5222, which is between 0.5 and 0.85. The release process mainly involves Fick diffusion and skeleton dissolution [33,34]. The nanofibers protect a small fraction of the drug that is preferentially released from the nanoparticles in a burst manner. Subsequent drug release from the fiber is accompanied by a slow onset of release from the tightly bound nanoparticles. This is a possible reason for the appearance of fick diffusion and skeletal dissolution combined release behavior [35].

### 3.5. Analysis of Cellular Uptake Characteristics In Vitro

The effective uptake by cells is an important step for drugs to exert their effect. The size and state of nanoparticles can affect their biological distribution, efficiency, and cell uptake behavior through adhesion and interaction with cells [36]. Figure 6a,b presents the particle size and potential distribution of the C6-NPs prepared in the experiment, similar to the prepared Cel-NPs. Therefore, the cellular uptake process of celecoxib can be observed qualitatively and quantitatively. The cytotoxicity test showed no cytotoxicity at the concentration of the blank nanocarrier solution at 125 µg/mL and 45 mg/mL nanofiber spinning solution (Figure 6c). The drug is electrostatically adsorbed with PLGA and is easy to ingest by endocytosis through the lipid bilayer of the cell membrane, and the drug is mainly released by simple diffusion in the cell [37]. As shown in Figure 6e, the two groups of preparations showed good uptake under the fluorescence microscope, and the probe could penetrate the cell membrane and enter the cytoplasm. Concerning the intake in the NPs-NFs group (30524 ± 8355.44) MFI was higher than that of the NPs group (11,727 ± 665.00) MFI at 0.5 h (*p* < 0.01) by flow cytometry. There was no significant difference in intake between the NPs and NPs-NFs groups at 4 h (*p* > 0.05) (Figure 6d). The lack of cell adhesion in the Cel-NPs solution may explain the rapid release of Cel-NPs in vitro but low uptake by cells. NPs-NFs are beneficial to macrophages in reducing the effective uptake time. The high clearance rate of macrophages in an inflammatory environment will lead to delayed recognition and post-processing of NPs, and well-designed carrier NPs can significantly avoid this recognition system and improve its effective delivery and long-term uptake in a delayed time [38]. Experimental results show that the delivery mode of NPs-NFs can be potentially applied to this mechanism of delayed identification and post-processing, but NPs alone do not have this advantage [39].

### 3.6. In Vivo Assessment of Cel-NPs-NFs Safety and Effectiveness

The rat hind limb grip measurement (Figure 7a) and behavioral scores (Figure 7b) showed that the rat arthritis model could be successfully established by intra-articular injection of sodium iodoacetate. As shown in Figure 7(ei), in the normal group, the color of the articular cartilage was uniform and white, and the surface was smooth and smooth without wear. In the model group (Figure 7(eii)), the surface of cartilage was rough and ulcerated, and the cartilage defect was so severe that the subchondral bone was exposed, and osteophytes were formed around it. The performance of the sham operation group (Figure 7(eiii)) was similar to the model group, with no aggravation or relief. In the positive control group (Figure 7(eiv)), the articular cartilage showed uneven ulceration, increased joint effusion, and cartilage loss. In the Cel-NPs-NFs group (Figure 7(ev)), the marginal line of cartilage was rough and not smooth, with increased joint effusion, but the joint remained relatively intact, with less cartilage loss. The results of the Pelletier score standard (Figure 7c) showed significant differences between the positive control (*p* < 0.05) and the Cel-NPs-NFs treatment groups (*p* < 0.001) compared with the model group. 

Hematoxylin–eosin staining showed that in the normal group, the cartilage structure was intact (Figure 7(fi)), the thickness of cartilage was normal, the surface layer was smooth without cracks, the cells were arranged in an orderly calcified layer, and there was no proliferation of subchondral bone. In the model (Figure 7(fii)) and sham operation (Figure 7(fiii)) groups, the cartilage became thin, and the fissure reached the calcified layer of cartilage. In addition, the lighter staining of the cartilage nucleus indicated that the cells were exfoliated and necrotic, the surface fibers exhibited proliferation, and part of the calcified layer was lost. Under the action of the chondrocyte compensation mechanism, the subchondral bone was invaded by blood vessels to some extent. In the positive control group (Figure 7(fiv)), the cartilage was slightly thicker than that in the normal group, the rough cracks on the cartilage surface reached the radiation layer, and the disordered arrangement of cells showed aggregated distribution. In the Cel-NPs-NFs treatment group (Figure 7(fv)), the thickness of the cartilage was normal, and there were cracks on the cartilage surface, reaching the transition layer. Compared with the model group, the structural integrity of the joint was maintained, the chondrocytes were slightly sparse and irregularly arranged, and the calcified layer of cartilage was difficult to distinguish from the radiation layer. Saffron fast green staining showed that in the normal section (Figure 7(gi)), the cartilage structure was intact and evenly stained, and the boundary between hyaline cartilage and calcified cartilage was a rugged comb-like structure. In the model and sham operation groups (Figure 7(g(ii,iii))), the cartilage structure was lost due to cartilage damage and the the disordered arrangement of cells. In the positive control group (Figure 7(giv)), the structure of cartilage was relatively intact, the boundary of the calcified layer was slightly blurred, the thickness of cartilage increased slightly, the chondrocytes were irregularly arranged, and the cartilage was stained lighter; therefore, the therapeutic effect was more pronounced. In the Cel-NPs-NFs treatment group (Figure 7(gv)), the cartilage structure was intact, the cartilage staining was slightly lost, the boundary of the calcified layer was slightly blurred, and the therapeutic effect was evident. According to the Mankin scores (Figure 7d), there was a significant difference between the positive control and model groups (*p* < 0.0001), and there was a more significant difference in the Cel-NPs-NFs treatment group (*p* < 0.05). The overall results showed that the significant difference in the Cel-NPs-NFs treatment group was more robust. A more likely reason may be that the administration mode of the Cel-NPs-NFs treatment group reached the cartilage surface directly without going through the systemic circulation, NPs could be retained on its surface for a long time because of its nature, and the NFs carrier also had a good retention lubrication effect. On the one hand, it could form a drug store in the joint cavity and release NPs for a long time, effectively inhibiting the cartilage damage caused by inflammation. Its lubrication effect could reduce the cartilage friction caused by its daily activities, with a double protective effect.

## 4. Conclusions

Using PVA as the spinning material, Cel-NPs-NFs were successfully prepared by loading stable NPs on the fiber membrane by electrospinning. Celecoxib was found on the fiber membranes in an amorphous state, with good resistance and hydrophilicity. The in vitro release test showed that its release ability was sustained for seven days due to the secondary encapsulation in the fiber membrane, and its release behavior changed from Fick diffusion to the joint action of Fick diffusion and skeleton dissolution. The in vitro uptake experiment showed that compared with the NPs and NPs-NFs systems, the cell uptake efficiency significantly improved in a short time. The in vitro experiments showed that the NPs-NFs system was beneficial in prolonging the drug release time and improving the cellular uptake efficiency. Furthermore, in the study of the rat arthritis model, the gross cartilage and hematoxylin–eosin staining after treatment showed that Cel-NPs-NFs had a better therapeutic effect than oral administration. To summarize, NPs-loaded NFs as a drug delivery system have great potential in the field of controlled drug release research.

## Figures and Tables

**Figure 1 pharmaceutics-15-00730-f001:**
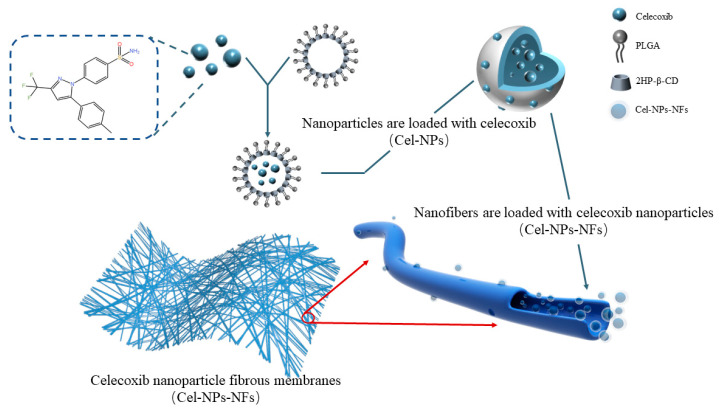
Schematic diagram of the mechanism of Cel-NPs-NFs preparation.

**Figure 2 pharmaceutics-15-00730-f002:**
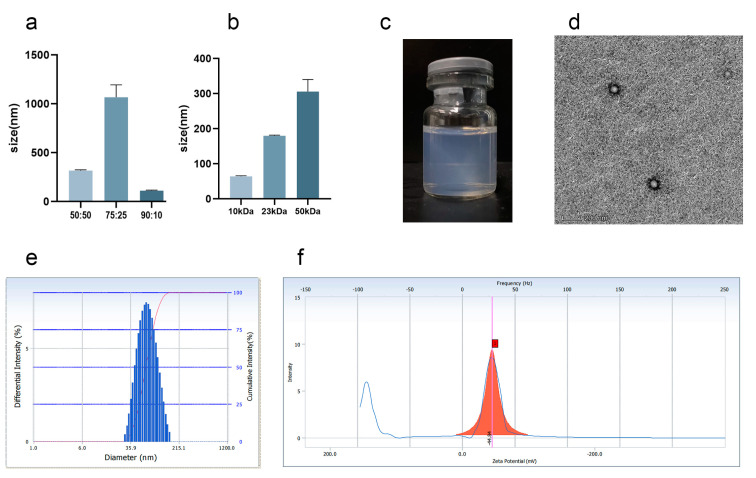
The preparation of Cel-NPs. (**a**) When the molecular weight of PLGA was 1 w, the particle size of Cel-NPs was 90% under the copolymerization ratio of 50:50, 75:25, and 90:10; (**b**) The average particle size of Cel-NPs with molecular weights of 10, 23, and 50 kDa at a PLGA copolymerization ratio of 90:10; (**c**) Cel-NPs external view; (**d**) Cel-NPs under a transmission electron microscope (scale: 200 nm); (**e**) The particle size distribution of Cel-NPs was prepared using PLGA with a molecular weight of 10 kDa and a copolymerization ratio of 90:10 as a carrier; (**f**) Potential distribution map (The red square on the x-axis is the potential value).

**Figure 3 pharmaceutics-15-00730-f003:**
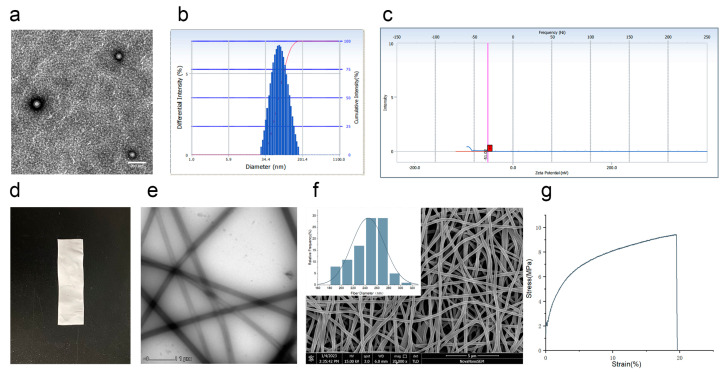
The characteristics of Cel-NPs-NFs; (**a**) Cel-NPs in nanofibers under a transmission electron microscope (scale: 200 nm); (**b**) Particle size distribution of Cel-NPs in nanofibers; (**c**) Potential distribution diagram of Cel-NPs in nanofibers (The red square on the x-axis is the potential value); (**d**) Cel-NPs-NFs appearance diagram; (**e**) Cel-NPs-NFs under a transmission electron microscope (scale: 1 µm); (**f**) Cel-NPs-NFs under a scanning electron microscope and diameter distribution of 100 fibers (scale: 5 µm); (**g**) Stress–strain curve of Cel-NPs-NFs.

**Figure 4 pharmaceutics-15-00730-f004:**
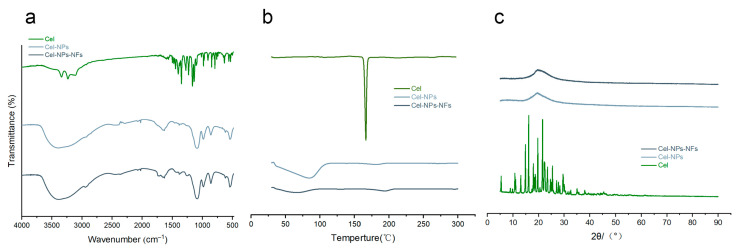
Discussion on the interaction mode of drugs and carriers IR, DSC, and X-RD; (**a**) Infrared characteristic map; (**b**) DSC curve; (**c**) X-ray diffraction pattern. (Cel, Cel-NPs and Cel-NPs-NFs).

**Figure 5 pharmaceutics-15-00730-f005:**
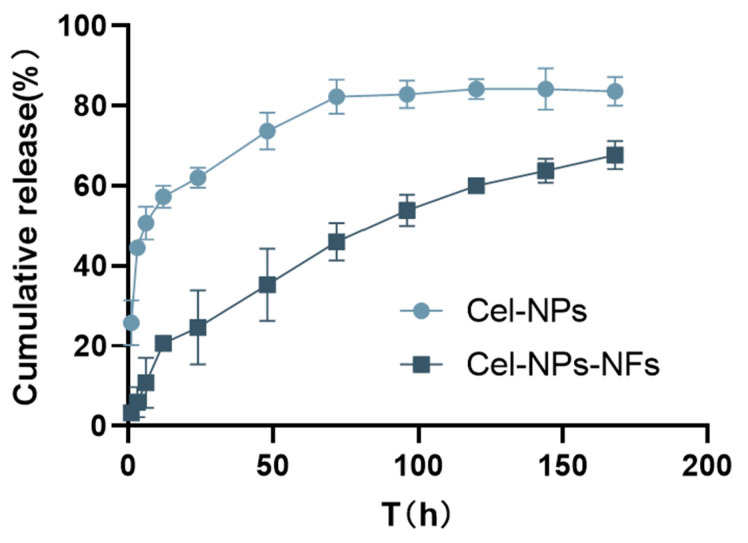
In vitro release curve of Cel-NPs and Cel-NPs-NFs in 168 h.

**Figure 6 pharmaceutics-15-00730-f006:**
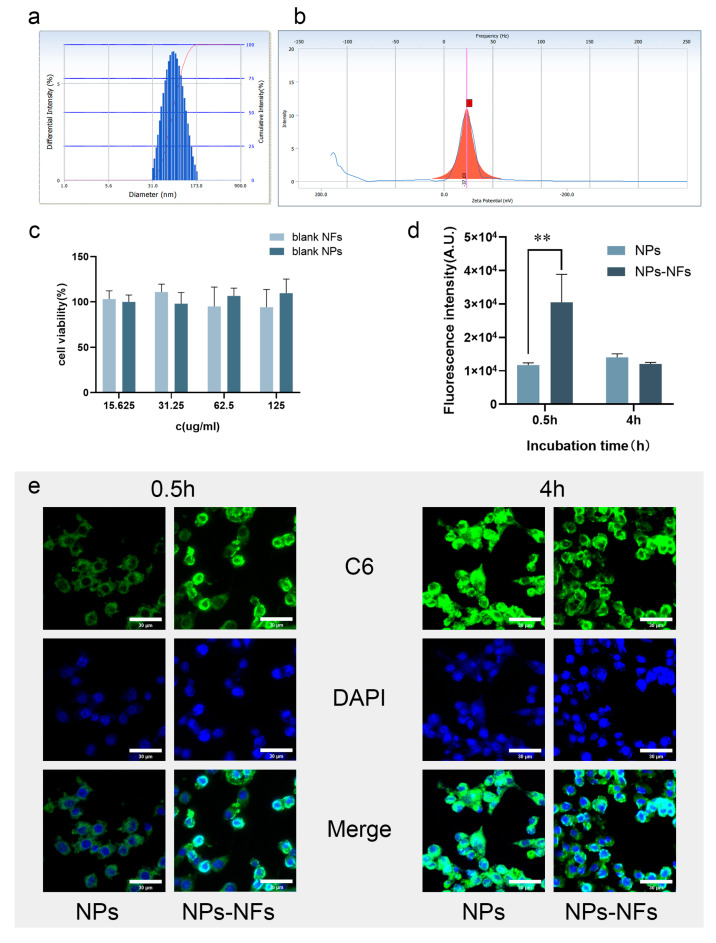
Uptake of RAW264.7 cells by fluorescence confocal microscopy and flow cytometry; (**a**) Particle size distribution of C6-NPs; (**b**) Potential distribution of C6-NPs (The red square on the x-axis is the potential value.); (**c**) Cytotoxicity study of blank preparation CCK8; (**d**) Fluorescence uptake analysis of flow cytometry at 0.5 h (** means *p* < 0.01) and 4 h; (**e**) Observation of cell uptake at 0.5 h and 4 h under confocal fluorescence (scale: 30 µm); C6 green fluorescence indicates the state of uptake instead of the model drug as a probe. The DAPI-stained nuclei are blue. Merge indicates the whole uptake state.

**Figure 7 pharmaceutics-15-00730-f007:**
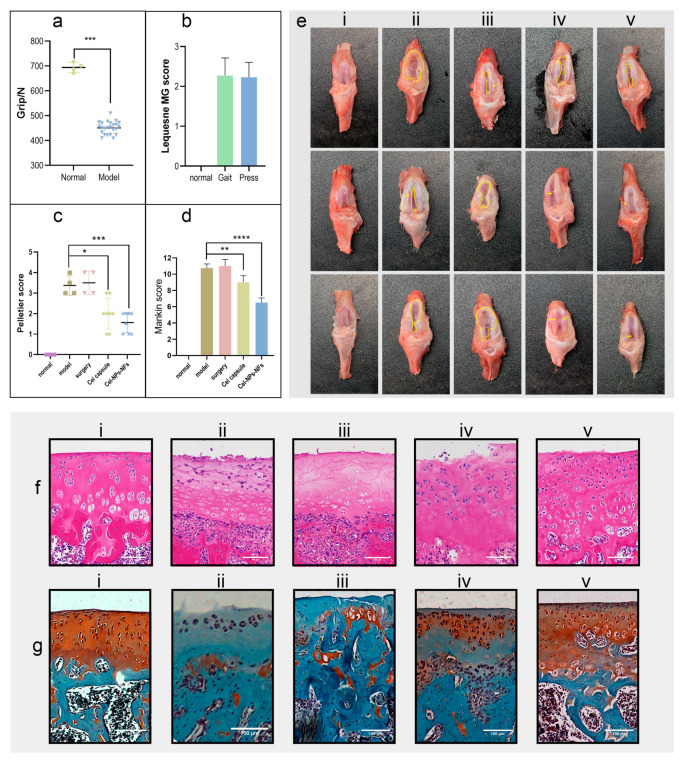
Study of the drug delivery performance in the OA rat model; (**a**) Analysis of hind limb grip of rats before administrating one-way ANOVA between the normally fed and model groups (*p* < 0.001); (**b**) The behavioral Lequesne MG score of OA model rats (including gait and knee joint compressions); 0 indicates the slightest degree without pathological changes, and 4 indicates the most severe pathological changes; (**c**) After treatment, the gross Pelletier score of the knee joint (corresponding to (**e**)), 0 was the slightest without lesion, 5 was the most serious, with Dunnett multiple comparisons (* means *p* < 0.05, *** means *p* < 0.001); (**d**) Combined with the HE section and saffron fast green staining section ((corresponding to (**f**,**g**)), according to the degree of pathological changes of Mankin score. A 0 score was the lightest without pathological changes, with 15 indicating that the lesion was the most serious, with the Dunnett multiple comparisons (** means *p* < 0.01, **** means *p* < 0.0001); (**e**) Gross view of cartilage (i, the normal group; ii, the model group; iii, the model sham operation group; iv, positive control group; v, the Cel-NPs-NFs treatment group); (**f**) HE section (scale: 100 µm); (**g**) Fuchsin fast green staining section (scale: 100 µm).

**Table 1 pharmaceutics-15-00730-t001:** Drug loading and encapsulation efficiency of different preparations.

Preparation	Drug Loading	Drug Loading Efficiency (%)	Encapsulation Efficiency of Nanoparticles (%)
Cel-NPs	422.28 ± 16.90 µg/mL	1.11 ± 0.04	88.23 ± 4.16
Cel-NPs-NFs	74.73 ± 0.99 µg/cm^2^	0.374 ± 0.005	

**Table 2 pharmaceutics-15-00730-t002:** Fitting of in vitro release model.

Preparation	Release Model	Fitting Equation	R^2^
**Cel-NPs**	Zero-order kinetics	Q = 0.2753t + 49.1036	0.6725
	First-order kinetics	Q = 77.8053(1 − $e^{-0.2023t}$)	0.7692
	Higuchi	Q = 4.2911$t^{\frac{1}{2}}$+ 37.3436	0.8504
	Ritger-Peppas	Q = 34.8465$t^{0.1841}$	0.9491
**Cel-NPs-NFs**	Zero-order kinetics	Q = 0.3816t + 11.5753	0.9319
	First-order kinetics	Q = 69.3391(1 − $e^{-0.0169t}$)	0.9767
	Higuchi	Q = 5.5293$t^{\frac{1}{2}}$ − 1.8841	0.9929
	Ritger-Peppas	Q = 4.8077$t^{0.5222}$	0.9937

## Data Availability

Data is not available due to privacy reason.

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
