# Peer review of "Formation of Hydrophilic Nanofibers from Nanostructural Design in the Co-Encapsulation of Celecoxib through Electrospinning"

_pharmaceutics, 2023, doi:10.3390/pharmaceutics15030730_

Round 1
Reviewer 1 Report
The manuscript presents the one-step co-encapsulation of PLGA nanoparticles in PVA hydrophilic nanofibers for therapeutic effect of osteoarthritis (OA). The manuscript can be accepted after several major issues are addressed:
- The aim of the research is not clearly defined in the abstract.
- Please extend the Introduction section.
-Which is the structure of a nanoparticle fiber membrane (Cel-NPs-NFs)? Please provide and schematic structure.
- Line 86-87: please modify - hydrolysis degree 88%
- Please convent the molecular weight of 1 w - in Da; kDa or g/mole
- The morphological investigation does not reveal the nanoparticles size within membrane. How the DLS analysis was done for nanoparticles entrapped into a membrane? Nanoparticle have the capacity to be somehow release from membrane? If yes, please provide the investigation.
- Please explain the interaction between the PLGA nanoparticles and PVA membrane: are the nanoparticles entrapped inside fibers or outside? Why?
- Line 373 The drug modified by PLGA is easy to ingest - Are there any chemical reaction between drug and PLGA? If yes, please describe the chemical reaction. If not, please rephrase the text.
Reviewer 2 Report
The paper presented herein is one of great significance and reasonable quality. The paper is well thought out. I highlight my specofoc concerns below
· The paper needs extensive grammar work throughout. There are clumsy sentences that need to be fixed.
· Line 49, the sentence needs to be rephrased.
· Line 85, Is that supposed to be PVA?
· Line 89, Spinning speed to mean injection rate?
· The methods are not well written. They need an overhaul. The section 2.6 needs to be described as separate parts. Individual information on how the DSC, XRD and FTIR are required. Was the ATR-FTIR used? There needs to be section describing the HPLC method. This was used to determine drug release and %EE. The statistical analysis section needs to be rewritten.
· Line 141 remove the
· Equation 2 needs to be further explained as the meanings are not clear to the reader.
· The results and discussion need more work. The authors assert that the encapsulation is confirmed by disappearance of peaks. FTIR has to detect the peaks associated with the API. It would be advisable for the authors to identify a peak unique to the API and use it as a marker peak for the determination of API encapsulation using a chemometric analysis. Furthermore, the drug release is not explained correctly or succinctly. Does the API have to diffuse from the NP and into the fibre prior to release? The NP loaded NF should have a double release mechanism which the authors do not dwell to explain. It would be beneficial to explain this phenomenon.
Reviewer 3 Report
In this manuscript, the authors present a one-step method to encapsulate celecoxib loaded PLGA nanoparticles within hydrophilic electrospun nanofibers. In general, the authors validate their hypothesis, however, some major concerns should be fully addressed prior to acceptance.
1. A proof reading needed for this manuscript. Their many typos throughout the whole manuscript. (e.g. , Page 1, line 32, “ increased obesity”, Page 2, line 85, “PVA as a membrane material”, Page 4 Line 188 etc.)
2. In Figure 1 d, the enlarged TEM figures should be provided. As the small size of PLGA nanoparticles (less than 100 nm), nothing could be observed at this magnification.
3. In Figure 2 b, the experiment details to determine the size of Cel-NPs in nanofibers should be added in materials and methods.
4. In Figure 2e,f,g, the control group (nanofibers without nanoparticles are all missing)
5. In Page 5 line 341, the caption of table 2 is mismatching.
6. The Table 2 is confusing and misunderstanding, the entrapment efficiency could be either the drug loading efficiency or the nanoparticle encapsulate efficiency.
7. In Figure 5c, the cell viability unit is percent, so the viability scale should from 0-100.
8. The Cel-NPs exhibited faster in vitro release than Cel-NPs-NFs, however, the cells revealed a rapid Cel-NPs-NFs uptake rather than Cel-NPs. The author should discuss the abnormal observation further.
9. Please keep the consistency of reference format (e.g. the capital letter of journal name).
Reviewer 4 Report
The subject of the manuscript is interesting, the study was properly designed, and the results are promising. However, the text needs improvement. Sometimes it was hard to follow the content.
Specific comments are given below:
The abstract seems incoherent. It contains doubled sentences - lines 18-19 and 22-23.
Line 28 – correct the verb form
Lines 49-51 - Correct the sentence so that it is referred to NSAID or celecoxib
Lines 57-59 – Correct the sentence.
Line 71 – place „solution” instead of „juices”
Lines 82-83 – combine two sentences to avoid the second „finally”
Line 85 – correct PAV
Lines 92-92 – „and finally cut to a 3×5 mm drug-containing film (Cel-NPs-NFs).” – It should be more precise: 3×5 mm film was cut from the obtained drug-containing membrane…
Line 102 – „performed” or „carried out” instead of „observed”
Line 104 – delete „small”
Line 106 – delete „naturally”
Lines 107-109 – correct the sentence
Line 132 – „the filtrate was filtered”?
Line 137 – Is „discrimination” correct? Shouldn’t it be „determination”?
Sections 2.5. and 2.8. There is no need to describe HPLC conditions twice.
Line 191 – the authors should explain why coumarin was selected for the cellular uptake study
Line 196 – what does „the concentration of celecoxib was 125 µg/mL gradient” mean?
Line 198 – use superscript
Line 339 – correct the Table description
Line 304 and 340 - the authors use „entrapment efficiency” and describe „encapsulation efficiency” in section 2.5 – terminology should be uniformed
Table 2 – correct the superscript of the Ritger-Peppas equation (Cel-NPs)
Line 367 – section description should be improved
Line 369 – correct „biology ical”
Line 417 – correct the verb form („bacome”)
Line 363 and 472 – what does „skeleton” mean?
Round 2
Reviewer 1 Report
Dear authors,
The manuscript can be accepted in the revised form.
Reviewer 2 Report
Can be published in present form
Reviewer 3 Report
The authors have address all my concerns and questions. No further comments with the revised manuscript.
Reviewer 4 Report
I recommend the manuscript for publication.